# Biomimetic Bacterial Membrane Vesicles for Drug Delivery Applications

**DOI:** 10.3390/pharmaceutics13091430

**Published:** 2021-09-09

**Authors:** Sajid Fazal, Ruda Lee

**Affiliations:** International Research Organization for Advanced Science and Technology, Kumamoto University, 2-39-1 Kurokami, Chuo-ku, Kumamoto 860-8555, Japan; f.r.sajid@gmail.com

**Keywords:** biomimetics, bacterial membrane vesicles, nanoparticles, drug delivery, antibiotic therapy

## Abstract

Numerous factors need to be considered to develop a nanodrug delivery system that is biocompatible, non-toxic, easy to synthesize, cost-effective, and feasible for scale up over and above their therapeutic efficacy. With regards to this, worldwide, exosomes, which are nano-sized vesicles obtained from mammalian cells, are being explored as a biomimetic drug delivery system that has superior biocompatibility and high translational capability. However, the economics of undertaking large-scale mammalian culture to derive exosomal vesicles for translation seems to be challenging and unfeasible. Recently, Bacterial Membrane Vesicles (BMVs) derived from bacteria are being explored as a viable alternative as biomimetic drug delivery systems that can be manufactured relatively easily at much lower costs at a large scale. Until now, BMVs have been investigated extensively as successful immunomodulating agents, but their capability as drug delivery systems remains to be explored in detail. In this review, the use of BMVs as suitable cargo delivery vehicles is discussed with focus on their use for in vivo treatment of cancer and bacterial infections reported thus far. Additionally, the different types of BMVs, factors affecting their synthesis and different cargo loading techniques used in BMVs are also discussed.

## 1. Introduction

For developing translatable engineered nanomedical systems for therapeutic and diagnostic applications, it is essential to consider the various different engineering and biological roadblocks these would encounter on the path to translation. From the engineering standpoint, large scale uniform production of nanoparticle (NP) systems is difficult to achieve primarily because of the complexity in their design [1,2], which leads to manufacturing difficulties in scale-up, quality control issues, downstream purification complexities, and increased cost of production. While from the biological standpoint, the immunogenicity of the NP system as a whole as well as its individual components, nanotoxicity, and overall therapeutic efficacy can further hinder its translatability [3,4] (Figure 1). In the past decade, there has been a collective effort towards addressing these issues, especially with concerns regarding the immunogenicity and biocompatibility of the designed NPs. Specifically, there has been a rise in research related to the development of NP systems that partly resemble or mimic non-immunogenic biological entities called as biomimetic/bioinspired NPs [5,6,7]. Such biomimetic/bioinspired NPs are considered to not only be non-immunogenic with reduced toxicity, but also possess superior pharmacokinetic properties [8,9,10] (due to lower macrophage clearance), paving the way for extensive research regarding their use for various therapeutic and imaging applications.

These biomimetic/bioinspired nanoparticles are considered to possess several advantages as compared to conventional NP systems composed of polymeric and inorganic materials. The most important advantage that they provide is their high biodegradability and non-toxic degradation products that can be easily cleared from the body without eliciting any long-term toxicity and bioaccumulation effects. Additionally, as the building blocks for the synthesis of such NPs are biomolecules, they can be harnessed directly from biological sources, which in some cases (e.g., albumin) can be obtained in large scale at low cost. Biomimetic/bioinspired NPs have been reported to be synthesized through a number of different materials and approaches which include (a) nanoparticles derived directly from biomolecules, (b) biomaterial coated nanoparticles, and (c) cell membrane-coated nanoparticles (Table 1, Figure 2). Nanoparticles that are derived directly from biomolecules are designed using a bottom to top synthesis strategy wherein biologically derived components such as albumin [11], casein [12,13], starch [14,15], gelatin [16,17], etc. are used directly to engineer and assemble the NP system that is loaded with a cargo of interest. Of note, the FDA-approved nanomedicine Abraxane^®^, which consists of the chemotherapeutic drug paclitaxel bound to albumin [18], is a prime example of this type of NP. Additionally, these biomolecules have also been used for coating NP surfaces in order to combine their biocompatibility with the desired functional ability of engineered NPs [19,20]. This strategy is particularly important since the pharmacokinetic property of a drug delivery system is primarily dependent on its surface physicochemical properties. By appropriately coating a functional polymeric/inorganic NP system with suitable biomolecules (Table 1), its interaction with different blood components in the body (including proteins and macrophages) is favorably altered so as to impart improved circulation time, higher bioavailability, and reduced clearance. Another widely used approach that has gained widespread attention involves the surface modification of synthetic NPs with cell membrane components [21] obtained from cells such as erythrocytes [22,23,24], platelets [25,26,27], macrophages [28,29,30], etc. This leads to the creation of a phospholipid bilayer surface on the NPs, thus affording them advantageous properties similar to that of liposomal structures. This strategy can be considered to be superior as compared to biomolecule coating, since the transfer of cell membrane components and the corresponding functional proteins (including cell membrane receptors, signaling proteins, etc.) provide added functionality to the NP system as a whole such as cancer cell targeting, immune evasion, biobarrier penetration, etc.

In contrast to the above methods of synthesis of biomimetic/bioinspired NPs, there is another approach that has recently gained widespread attention in the area of nanomedicine research. This involves the direct use of naturally synthesized extracellular vesicles [40] that are ubiquitously found to be produced by all cells (Figure 2D). Among the different types of extracellular vesicles such as microvesicles, exosomes and apoptotic bodies, the nano-sized extracellular vesicles called ‘exosomes’ obtained either from mammalian cells [41,42,43] or ‘Bacterial Membrane Vesicles (BMV)’ obtained from bacteria [44,45], have been demonstrated as excellent drug delivery agents particularly owing to their nanoscale size. Unlike other conventional nanoparticles that could have a solid core structure, the nanovesicle structures have a hollow hydrophilic interior (similar to liposomes) and can be used to transport drugs and other cargo. These naturally derived vesicles will exhibit a surface chemical composition identical to the parent cell from which they are obtained, and therefore would demonstrate high biocompatibility and low immunogenicity. Such favorable characteristics make them ideally suited for easy translation from a biological standpoint. Moreover, as they are biologically synthesized by cells directly, no chemical synthesis step would be required for their production and they can be produced by optimizing the cell culture and growth condition.

Even though such biomimetic nanovesicles could make it easier to scale the biological roadblock of clinical translation, it is important to overcome the large-scale design and engineering roadblocks with regards to their synthesis. Taking this into consideration, exosomes derived from mammalian cells would be particularly difficult to translate due to the challenge involved in undertaking large-scale mammalian cell culture to obtain exosomes in large quantities, in addition to the high cost for maintaining mammalian cell culture conditions at an industrial scale [46]. In this regard, exosomes obtained from bacterial sources, i.e., BMVs, could have high translation potential. The mass production of bacteria in bacterial growth tanks would be relatively easier to accomplish and the subsequent associated costs would also be relatively low [47], as compared to mammalian cell cultures. Additionally, a unique advantage associated with utilizing bacteria for BMV synthesis is their ease of genetic engineering which can help specifically design and produce BMVs with functional moieties [48]. In this review, the biomedical applications of BMVs are discussed with regards to their use as drug delivery vehicles for cancer therapy and antibacterial therapy. Particularly, attention will be focused on the bacterial sources for BMV production, their separation and purification, characterization techniques, drug loading strategies, and their in vivo biomedical applications reported thus far.

## 2. Types of BMVs and Factors Affecting Their Synthesis

Broadly, two different types of BMVs can be considered to exist based on the Gram staining of the bacterial source from which they are produced. BMVs that are secreted from Gram-negative bacteria are generally termed Outer Membrane Vesicles (OMVs), while those that are secreted from Gram-positive bacteria are simply called membrane vesicles (MVs) or Cytoplasmic Membrane Vesicles (CMVs). The reason the BMVs secreted from Gram-negative bacteria are called OMVs is because they originate from the outer membrane of the complex cell envelope [49] that encompasses the Gram-negative bacteria (Figure 3). On the other hand, MVs synthesized from Gram-positive bacteria originate directly from the cytoplasmic membrane of the simple Gram-positive bacterial cell wall [50]. Apart from these typical BMVs, several other structures such as Outer–Inner Membrane vesicles [51], Explosive Outer Membrane Vesicles [52], and Tube-shaped Membranous structures [53] have also been identified to be secreted by bacterial cells.

Even though the route of synthesis of BMVs is not clearly understood, several hypotheses for the same can be made by evaluating their composition and relative concentrations of protein, lipids, and nucleic acid contents [54]. As OMVs contain lipids and proteins that are typically present in the outer membrane and periplasmic space of their parent source bacteria, they are considered to originate from the gram negative bacterial outer membrane through cell membrane blebbing mechanisms [55]. On the other hand, Outer–Inner Membrane vesicles, which consist of two membrane layers (from both the outer membrane and inner membrane), additionally contain cytoplasmic protein and DNA and are considered to originate from the inner membrane of Gram-negative bacteria, by pinching off cytoplasmic cell components [51,56,57]. Outer–Inner Membrane vesicles are sometimes also found to contain chromosomal DNA, in which case they are considered to originate due to explosive cell lysis that result in cell death [52,58]. The synthesis of OMVs in Gram-negative bacteria are thought to occur due to defects in the peptidoglycan layer of the bacterial cell wall which can lead to the dissociation of the outer membrane. These defects can arise due to a number of factors such as disrupted crosslinking between the peptidoglycan and the outer membrane [59,60], accumulation of misfolded proteins in the periplasmic space [61,62], and through ‘bilayer coupling’ effects that are brought about by molecules that induce membrane curvatures [63,64]. Another possible mechanism for vesicle blebbing involves the action of the endolysin/autolysin enzyme, that can degrade the peptidoglycan layer. In Gram-negative bacteria, the action of endolysin leads to membrane instability that causes explosive cell lysis and eventual vesiculation [52,65]. For Gram-positive bacteria, the cytoplasmic membrane vesicles are considered to arise either from dying cells [65] or from other conservative blebbing mechanisms [56]. Here, the action of endolysin/autolysin is also considered to play a key role in the formation of CMVs. However, as Gram-positive bacteria have a thicker peptidoglycan layer, membrane instability effects are relatively less pronounced which led to the protrusion of the cytoplasmic membrane through pores in the peptidoglycan layer eventually leading to the release of CMVs.

A number of different genetic and environmental factors can affect vesicle formation in bacteria. Genetic factors can predispose a bacterium to produce more vesicles due to the accumulation of misfolded proteins in the peptidoglycan layer or due to lipid and protein composition in the outer membrane that can affect membrane curvature or cell envelope cross-linking. An example of a hypervesiculating bacteria is the *Escherichia coli* JC8031 produced by the genetic knockout of tolRA gene, that leads to membrane instability in the *E. coli* cell envelope [66]. Due to this hypervesiculating nature, *E. coli* JC8031 has been explored for various biomedical applications including the development of vaccines [67,68] and for drug delivery (discussed below). Environmental factors including bacterial growth conditions, medium composition, and other stress factors (including thermal stress [69,70] and antibiotic stress [71]) can also increase the release of BMVs.

BMVs play important roles including intracellular communication such as horizontal gene transfer between different bacterial species [72,73], immunomodulation in a potential host [74], aiding the formation of bacterial biofilms [75], and many others. In the human body, BMVs not only play important role in bacterial infection, but also play a protective role in preventing inflammation such as from commensal bacteria that reside in the gut [76]. Information regarding their structural composition, functions and mechanism of action are not yet fully unraveled, however readers are referred to exhaustive resources [77,78] that provide up-to date knowledge regarding these.

## 3. BMV Source for Cargo Delivery

BMVs for cargo delivery have been reported to be procured from a wide variety of different Gram-negative (*Escherichia coli* [79], *Acinetobacter baumannii* [71], *Cystobacter velatus* [80], *Klebsiella pneumoniae* [81], *Salmonella typhimurium* [82], and *Salmonella enterica* [83]) and Gram-positive bacteria (*Staphylococcus Aureus* [84] and *Lactobacillus acidophilus* [83]) sources which have been demonstrated for use in various biomedical applications (Table 2). Unlike other biomimetic NPs such as exosomes [41] or erythrocyte membrane mimicking NPs [23], BMVs generated from bacteria have the potential to produce an immunogenic response in vivo due to the presence of LPS (Gram-negative) or LTA (Gram-positive) and other non-human bacterial proteins on the BMV surface [85]. To counter this effect or to increase their tolerability, attenuated bacterial strains such as the *E. coli* K-12 W3110 strain [86] (carrying an *msbB* mutation which produces under-acylated LPS) or the attenuated *S. Typhimurium* strains [82] have been reported, which can exhibit reduced endotoxicity when used in vivo. However, the immunogenic potential of BMVs from non-attenuated bacterial strains have also been reported successfully for different applications including immunotherapy [87] and antibiotic delivery [71]. Additionally, BMVs have also been obtained from genetically engineered bacteria such that the synthesized BMVs have surface modifications that impart specific functions (IgG [88] and anti-HER2 affibody [86]) or are pre-loaded with cargo such as enzymes (luciferase [89] and phosphotriesterase [90]) or a therapeutic agent (melanin [91]). For successful translation, however, it is imperative that BMVs be either generated in large quantities using bacterial bioreactors or through the use of hypersecreting BMV strains such as the *E. coli* JC8031 [89]. Additionally, to circumvent issues related to toxicity and safety during clinical translation, utilizing non-pathogenic commensal bacteria such as *Bacteroides thetaiotaomicron* (that are part of the intestinal microbiota) could also be a viable alternative [92].

## 4. Separation, Purification, and Storage of BMVs

The separation and purification of BMVs from bacterial culture typically involves the following steps (Figure 4): (1) Centrifugation—employed at low speeds for the separation of bacteria from the culture suspension; (2) filtration and concentration—using 0.45 µm filters followed by concentration of crude BMVs using a 100 KDa membrane; and finally, (3) ultracentrifugation at reduced temperatures. Additional steps involving multiple filtration and centrifugation steps are routinely employed to separate the BMVs. Caution on multiple centrifugation steps must be taken as excessive centrifugal forces may disrupt the vesicular structure of the BMVs or lead to vesicle clumping.

For the purification of BMVs from extraneous proteins purification steps such as sucrose density gradient centrifugation and size exclusion chromatography techniques are utilized. However, these purification steps can affect total BMV yield, as BMVs have a wide size distribution and composition that can affect their density leading to reduction in BMV content in the final yield. Additionally, to remove the presence of free endotoxins, BMV concentrates could be purified using endotoxin removing columns.

For long-term storage of BMVs, most studies have employed low temperature storage condition of −80 °C wherein the BMVs themselves are resuspended either in PBS or water with or without anti-freeze compounds such as glycerol. However, one report which assessed the storage stability of BMVs at 4 °C, −20 °C, −80 °C, and lyophilized powder conditions (for storage between 7 and 75 days) has found that compared to the low temperature storage conditions, lyophilization of BMVs produced the lowest reduction in BMV concentration and size [80]. It is therefore imperative that a universal storage protocol be developed for long-term BMV use without affecting their size, physico-chemical stability and surface protein activity.

## 5. BMV Characterization Techniques

The physicochemical characterization techniques that are routinely applied for nanoparticles are also used for BMV characterization and consists of DLS and Zeta potential measurements, NTA analysis and TEM imaging. Typically, BMVs are 30–300 nm in size and are negatively charged. Figure 5A shows the TEM images of OMVs obtained from both wild type *E. coli* and its Δ*msbB* mutant [83], and were found to have a similar hydrodynamic diameter of ~38 nm. However, upon comparing the yield of OMVs obtained from both strains through protein quantification, the Δ*msbB* mutant was found to produce significantly higher yield as compared to the wild type strain.

By undertaking such protein concentration measurements to quantify the yield of BMVs obtained after separation and purification, bacterial growth culture conditions can be optimized in order to improve BMV yield, as the external growth conditions and medium can significantly alter bacterial growth rate and the corresponding vesicle release. This was interestingly depicted for OMVs obtained from *A. baumannii* (Figure 5B) when grown in the presence of different antibiotics at sub-lethal concentrations [71]. Through protein content and particle number analysis (NTA), it was found that only in the presence of levofloxacin, ~2.47-fold increase in the number of OMVs were observed. Correspondingly under these conditions, a significant increase in protein yield and OMV hydrodynamic diameter was also observed as compared to treatments with other tested antibiotics and untreated control (discussed in detail below).

Apart from the above physicochemical characterizations, BMVs are also bio-chemically characterized to determine the specific proteins that are expressed on its surface which are derived from the bacterial parent source. This is particularly important to assess the biological activity of the synthesized BMVs. These surface expressed proteins can be utilized as an anchor point to selectively express other functional proteins of interest or to load a suitable cargo. In one report, the bacterial membrane protein α-pore-forming toxin Cytolysin A was used to anchor anti-HER2 affibody [86] on its surface, so as to impart active targeting capability towards cancer cells that overexpress HER2. Similarly, in another report, the bacterial membrane protein OmPA was used to selectively load an enzyme cargo in vitro within the lumen of synthesized BMVs [90] (discussed later).

## 6. Cargo Loading and Surface Modification Using BMVs

The loading of cargo into BMVs have been undertaken through many different active (energy dependent) and passive (energy independent) loading mechanisms as reported in literature (Figure 6, Table 1). Among the different active loading techniques discussed below, co-extrusion and sonication have also been employed for the direct surface modification of NPs. By Np surface modification, the bio-(physicochemical) property of the parent source bacteria can be transferred onto the NPs for harnessing their function as discussed in specific examples below.

### 6.1. Active Cargo Loading

#### 6.1.1. Electroporation

Electroporation is usually employed as a non-viral gene delivery technique in vitro and in vivo. The technique involves the application of short high-voltage pulses to cells to form pores within its cell membrane to create a transient state of permeability [94,95]. This state of permeability allows the entry of drugs and fluorochrome compounds and even large-sized cargo such as nucleotides, which is optimized by varying the electric pulse and its duration. If optimized correctly, the phospholipid membrane then recovers its structure once this process is completed, without incurring any irreversible damage. Being applicable to cells, particularly to cell surface membranes, this process has been naturally extended for cargo loading of other lipid bilayer structures which can act as delivery vehicles such as biomimetic exosomes [96,97,98].

The first report on the use of electroporation to load cargo into BMVs was published in 2014 wherein BMVs from Gram-negative *E. coli* was used for the delivery of therapeutic siRNA [86]. The loading of siRNA in this case was achieved through the electroporation technique (at 700 V and 50 µF) leading to a high loading efficiency of 15 wt% siRNA in BMVs (Figure 7A). Here, fluorescent dye-labeled siRNA was loaded into BMVs, which extended its usage as a theranostic agent for cancer. Apart from nucleotide loading, metallic gold (Au) NP has also been loaded successfully into BMVs [93] (Figure 7B). In this study, small-sized Au NPs (<10 nm) could be loaded into *P. aeruginosa* BMVs by applying an optimal voltage of 470 V and 1 pulse yielding an encapsulation efficiency of ~35%. Note here that when *P. aeruginosa* BMVs alone were subjected to a higher electroporation voltage of 1500 V, a reduction in their structural stability was observed, indicated by a reduction in its protein concentration and an overall increase in the standard deviation of its hydrodynamic diameter. Thus, it is important to optimize the parameters for electroporation in order to successfully load BMVs with a desired cargo. Nevertheless, such Au NP-loaded BMVs could have wide biomedical applications, and this process of electroporation could be used for the loading of other types of cargo such as iron-oxides for MR imaging applications or quantum dots for fluorescence imaging.

#### 6.1.2. Co-Extrusion/Surface Modification

Co-extrusion has been reported to be widely used as a technique for surface modification of NPs, wherein a material of interest is mixed with the NP and extruded together so as to force an interaction between them [99]. Specifically, a number of different biomimetic NPs have been synthesized through this route by coating NP surface with cell membrane structures such as those obtained from red blood cells [23], leukocytes [100], cancer cells [101], platelets [102], etc. The technique mostly involves the isolation of the cell surface membranes followed by repeated mechanical extrusion of these membranes with the NP of interest through polycarbonate membranes of varying pore sizes. More recently, this has been utilized for the synthesis of exosome membrane coated NPs [103,104,105] and has now been extended to bacterial membranes and BMVs as discussed below.

In one such study, 30 nm citrate stabilized Au NPs were surface modified (Figure 8(Ai) using *E. coli* derived BMVs (~30–30 nm diameter) by extruding their mixture 7 times through a 50 nm polycarbonate porous membrane resulting in the production of ~42 nm surface modified Au nanoparticles [87]. This extrusion and surface coating process resulted in a natural increase in the hydrodynamic diameter of the Au NPs which was caused by the presence of BMV proteins on its surface. To further confirm the presence of surface-bound proteins on Au NPs, a fluorescence quenching analysis was done using FITC-thiol fluorochrome, as Au NP surfaces are known to be ultra-efficient quenchers (Figure 8(Aii) [106]. Due to successful surface modification, fluorescence quenching was observed only when unmodified Au NPs were mixed with FITC-thiol. It was found that the surface-modified BMVs contained ~8 wt% surface proteins as measured through the BCA protein assay, and this led to an increase in the stability of Au NPs in a physiological environment when compared to unmodified bare Au NPs. Similar to the above study, the coating of BMVs on drug loaded micelles have also been demonstrated [82]. Here, two different steps of extrusion steps were carried out to create a unique BMV modified micelle structure. BMVs isolated from *S. typhimurium* were first modified to incorporate a polymer, PEG-RGD by extrusion through a 220 nm polycarbonate membrane. This was done initially to reduce the immunogenicity of BMVs and to impart active targeting capability to them. These modified BMVs were then further extruded with tegafur-loaded F127 micelles, to obtain BMV coated micelle structures (Figure 8B).

#### 6.1.3. Sonication/Surface Modification

Similar to the co-extrusion technique, sonication is a simpler alternative which can be used for surface modification of NPs. The application of ultrasonic frequencies to cell membrane components can lead to their disruption and subsequent attachment on the surface of NPs [22,107]. Such NPs have altered physico-chemical properties that mimic cell surface characteristics in vivo.

In another report, antibiotic-loaded polymeric PLGA NPs were coated with BMVs for investigating active delivery of antibiotics to infection sites in vivo [84]. Here, BMVs isolated from *S. aureus* and *E. coli* bacteria were coated on vancomycin and rifampicin loaded PLGA NPs by mixing the NPs and BMVs at a 2:1 mass ratio followed by bath sonication of the resultant mixture. The protein loading on the surface modified PLGA NPs were found to be ~7 wt%. An interesting observation that was observed in this study, was the specific uptake of BMV membrane coated PLGA NPs by macrophages that were previously infected with bacteria. Importantly, it was found that this uptake was dependent on the specific bacteria that the macrophage had previously been infected to, i.e., macrophages infected with *S. aureus* or *E. coli* bacteria specifically showed a significant uptake of PLGA NPs that were either coated with *S. aureus* or *E. coli* BMVs, respectively.

### 6.2. Passive Cargo Loading

#### 6.2.1. Simple Incubation

BMVs can also be loaded through a passive loading technique of simple incubation with cargo. In one such study, Gram-negative *K. pneumoniae* BMVs were loaded with chemotherapeutic drug doxorubicin-hydrochloride [81] by incubating the BMVs with the drug at 37 °C for 4 h, followed by the removal of free drug using 100 KDa membrane ultrafiltration and PBS wash repeatedly. The encapsulated drug in BMVs were quantified using mass spectrometry analysis which showed that maximum encapsulation efficiency of ~78% could be obtained when the drug and BMVs were incubated at a mass ratio of 1:45. In other reported studies, the simple incubation technique was used for the fluorescent labelling of BMVs for in-vivo imaging applications. [79,83] Here, the BMVs were incubated with an NIR dye Cy7 mono NHS ester for 2 h at 37 °C followed by their separation from excess dye through ultracentrifugation.

#### 6.2.2. Incubation with Parent Bacteria

A simple mode of loading BMVs is achieved through incubating the cargo material with the bacteria of interest during its growth phase. The bacteria in this case would engulf the cargo present in the extra cellular environment and sort the same into BMVs which are shed from the bacteria. In one such extensive study, antibiotic-loaded BMVs were synthesized from *A. baumanii* by culturing the bacteria in antibiotic containing medium [71]. Specifically, different antibiotics such as ceftriaxone, amikacin, azithromycin, ampicillin, levofloxacin, ciprofloxacin, and norfloxacin were added at different fractions of their respective Minimum Inhibitory Concentrations (1/2, 1/4, 1/8). Characterization studies post drug loading showed that the phospholipid wall in BMVs had thickened when they were loaded with ceftriaxone, amikacin, azithromycin, and levofloxacin. Additionally, it was found that the antibiotic levofloxacin at 1/8 minimum inhibitory concentration produced the highest encapsulation efficiency in the generated BMVs, with ~120 µg of Levofloxacin/10^12^ BMV particles, while, on the other hand, ciprofloxacin, azithromycin, and ampicillin antibiotics failed to be loaded into the secreted BMVs. The authors report that the reason for this wide difference and preferential loading of drugs into BMVs could be complex and further studies are therefore needed to understand the same. Interestingly, this study also reports that mere incubation of drug cargo with empty BMVs did not lead to any cargo loading. This phenomenon could suggest that the loading of antibiotics internally through bacteria in this case could have occurred through a drug efflux mechanism.

In another report with a similar objective of developing BMVs for anti-bacterial applications, BMVs were synthesized and isolated from non-pathogenic myxobacteria (soil bacteria) *C. velatus*, Sorangiineae species strain SBSr073 [80]. Here, the BMVs were studied directly for their antibacterial property since the myxobacterial species are known to be predatory towards other competing Gram-positive and Gram-negative bacteria while using them as a nutrient source.

#### 6.2.3. Transformation of Parent Bacteria

Cargo loading of BMVs can also be employed intrinsically by transformation of the parent bacteria (genetic engineering) wherein a plasmid expressing the desired protein cargo is engineered. In one such report, multifunctional BMVs were synthesized using genetically engineered *E. coli* [88,89]. To load cargo within BMV lumen, native proteins anchored to the periplasmic side of the outer membrane were utilized as an anchor point. Specifically, a bioluminescent agent, NanoLuciferase enzyme, was loaded within BMV lumen by anchoring it to SlyB protein by co-expressing them in a bacterial expression system. Further, a protein scaffold was used to anchor IgG antibody to BMV surface by binding to an BMV surface membrane expressed Ice Nucleation protein. The loading of NanoLuciferase within BMVs was confirmed by Western blotting analysis which showed that the cargo protein degraded upon the use proteinase K only when the BMVs were lysed using SDS. Such synthesized BMVs could be used for biosensing applications for the detection of any antigen. In another report, BMVs were utilized for the packaging of a bioremediating enzyme phosphotriesterase within BMV lumen. [90] For this, a SpyCatcher/SpyTag bioconjugation system was employed in *E. coli* bacteria, wherein the native surface membrane protein, OmpA was bound to the SpyCatcher domain while the phosphotriesterase enzyme was bound to the SpyTag domain. The packaging of cargo was attempted by co-transformation using expression vectors containing the appropriately modified SpyCatcher and SpyTag genes, which could lead to the formation of the hybrid protein within the bacteria, and its release in BMVs. It was found that the cargo packaging into BMVs increased its stability and robustness, possibly allowing its usage in harsh environmental conditions for bioremediation.

An interesting study on the use of BMVs for theranostic application was recently reported, wherein genetic engineering techniques were employed for loading a theranostic agent, melanin into *E. coli* generated BMVs [91]. Here, the enzyme tyrosinase, which is responsible for the production of melanin, was encoded into an expression vector. Upon the introduction of the tyrosin substrate, the tyrosinase enzyme catalyzed its conversion to melanin within bacterial cytosol and periplasmic space which could then be packaged into released BMVs (Figure 9). Thus, the versatility of genetic engineering techniques enables it to be employed for the loading of many different types of cargo.

## 7. Drug Delivery Applications of BMVs

The therapeutic application of BMVs has largely been explored pre-clinically for its use as an immune-modulating agent [108,109,110]. This is primarily because of the presence of antigenic protein molecules in BMVs which when used, may trigger a favorable immune response in the body. Until now this phenomenon has been successfully translated to clinics for the development of a vaccine against *Neisseria meningitidis* serogroup B (Bexsero^®^ developed by Novartis) [111]. In some cases, BMVs have been demonstrated to amplify the immunogenicity of a low immunogenic protein antigen by acting as a vaccine delivery system [112,113]. Additionally, the ease of genetic modification of the bacterial source has also contributed to it being utilized as an efficient and promising immunomodulator.

The use of BMVs for drug/cargo delivery applications has only been explored recently with only a handful of published literature reports. Most of the applications for BMVs has primarily been focused either on cancer therapy or antibacterial therapy (Figure 10).

### 7.1. Cancer Therapy

In one such report, BMVs isolated from non-pathogenic attenuated *K. pneumoniae* were utilized for the delivery of doxorubicin [81]. Anti-tumor studies carried out using such BMVs at a dose of 2 mg/Kg of DOX (injected intraperitoneally every day for 11 days) in A549 tumor bearing BALB/c nude mice showed a significant reduction in tumor volume as compared to the use of free drug, empty BMVs, and even doxorubicin-loaded liposomes (Figure 11A). In fact, it was observed that the rate of reduction in tumor volumes was found to be greater for DOX-loaded BMVs as compared to DOX-loaded liposomes, signifying that BMVs showed a better therapeutic response. This added therapeutic response observed in DOX-loaded BMVs could be attributed to the favorable immune response that BMVs can induce in vivo which in conjunction with chemotherapeutic drugs leads to generation of a higher therapeutic efficacy. This was supported by tumor volume reduction studies in the same report that showed that the use of bare BMVs alone in vivo lead to a significant reduction in tumor volume as compared to untreated controls. Further, it was also observed that there occurred a significant accumulation of murine macrophages in tumor tissues that were treated with both doxorubicin loaded BMVs and empty BMVs. Pharmacokinetic analysis showed that the use of drug loaded BMVs lead to a greater drug retention in tumors that lasted for longer periods of time as compared to DOX-loaded liposomes with a concurrently lower retention found in the heart (Figure 11B). As a result, the cardiac toxicity (which is notable in the use of doxorubicin) was found to be significantly reduced when DOX-loaded BMVs were used (as measured through analysis of lactate dehydrogenase, aspartate aminotransferase, and creatine kinase isoenzyme in blood), which were further supported through histopathological analysis of cardiac tissues. Overall, the pharmacokinetic profile of the loaded drug was improved (characterized by an increase in the drug half-life, reduction in clearance rate, and improved bioavailability) when BMVs were utilized as a drug delivery vehicle. Immunotoxicity analysis in C57BL/6 normal mice at the therapeutic dosage (over a period of 11 days) showed that the administration of both DOX-loaded BMVs and bare BMVs lead to a significant increase in serum cytokine levels which returned back to basal levels over a period of time. These results therefore showed that BMVs could be well tolerated in vivo and could be used as an effective drug delivery vehicle.

Similar to the above, a study depicting the use of *S. typhimurium* BMVs for combined drug delivery and immunotherapy against cancer was reported [82]. In vivo tumor therapy studies carried out using hybrid BMVs (BMV/micelle/drug) in B16F10 melanoma and 4T1 mammary tumor in C57BL/6 mice at a dose of 30 µg of BMVs (once/3 days for a total of 3 injections) lead to a significant reduction in tumor volume and increase in survival as compared to controls. Furthermore, this treatment also limited the spread of cancer metastatic nodules in lung tissues, which otherwise are prevalent in the B16F10 tumor model, which could explain the increase survival of mice observed on treatment. Interestingly, the synthesized BMVs also showed an immunoprotective effect against tumor. Mice that were pretreated with BMVs, when challenged with tumor cells later on, showed a delayed tumor growth response with significantly small tumor volumes (Figure 12). Even though the exact reasons for the same have not been elucidated in this report, these results show overall the promising effect that BMVs have towards developing a strategy for tumor prevention and treatment. Upon BMV administration, the in vivo cytokine analysis of blood and tumor samples showed that even though there occurred an increase in the cytokine levels of TNF-α, IFN-γ, IL-12, IL-4, and IL-17, the levels of these cytokine reduced to basal levels after 24 h. Overall, no blood toxicity and organ toxicity (including liver and renal functions) were found upon BMV administration.

Even though the above studies showed the ability of BMVs as a promising drug delivery agent, an interesting study has demonstrated its ability to be used as in its native form as a potential anti-tumor immunotherapeutic agent [83]. To demonstrate its applicability, unmodified BMVs isolated from both Gram-negative and -positive bacterial species were assessed for their ability to actuate an anti-tumor immune response in different tumor models in mice. Specifically, BMVs isolated from Gram-negative *E. coli* and *S. Enterica* and Gram-positive *S. aureus* and *L. acidophilus* were injected intravenously in BALB/c mice bearing CT26 colon adenocarcinoma at a 5 µg BMV dose (4 times at 3 days interval), significant tumor volume reductions were observed as compared to PBS injected controls (Figure 13A,B). Additionally, to demonstrate its diverse potential, BMVs from *E. coli* were assessed for their therapeutic response in CT26 colon adenocarcinoma and 4T1 mouse mammary tumor of BALB/c mice, and MC38 mouse colon adenocarcinoma and B16BL6 mouse melanoma cancer of C57BL/6 mice. At a 5 µg *E. coli* BMV dose injected intravenously (4 times at 3 days interval), significant tumor volume reductions were observed for all the treated tumor types. However, the reduction in tumor volumes in 4T1 and B16BL6 tumors were found to be less effective as compared to those observed in CT26 and MC38 tumors, which shows the important role tumor biology and characteristics have on the net therapeutic outcome. Interestingly, for the treatment of CT26 tumors, a long-term memory effect was observed for treatment using *E. coli* BMVs wherein secondary and tertiary challenges of CT-26 tumor cells were rejected in mice that recovered from the primary tumor challenge post BMV administration. These results show how BMVs have the ability to favorably modulate the immune system and possibly provide a protective environment to prevent tumor relapse as observed for other immunotherapeutic modalities [114]. Similar to other reports on the use of BMVs in vivo, an increase in the levels of cytokines and chemokines such as IL-12p40, IFN-γ, CXCL10, TNF-α, IL-6, and IL-12p70 were also observed in this study. However, it was observed that the cytokines CXCL10 and IFN-γ specifically showed elevated levels in the tumor tissues over 24 and 48 h which could imply that these cytokines play an important role in eliciting an anti-tumor immune response (Figure 13C,D).

Apart from utilizing BMVs for drug loading, BMVs have also been utilized for the delivery of therapeutic nucleic acids for the treatment of cancer. By loading siRNA targeting kinetic spindle protein, a protein essential for spindle formation and continuation of cell cycle, into BMVs obtained from Δ*msbB*
*E. coli*, in vivo treatment of liver cancer was demonstrated [86]. Significant reduction of HCC-1954 xenografts in nude mice was observed upon intravenous administration of siRNA-loaded BMVs at a 4 µg dose siRNA injected on alternate days over a 22-day treatment period as compared to controls. Serum cytokine analysis showed elevated levels of TNF alpha, IL6, and IFNγ in C57BL/6 mice upon repeated dosing (at 10–20 µg siRNA) over 4 consecutive days. However, this elevation was observed only for a brief period of 3 h post-administration and would return back to basal levels in 24 h. Note that lethal dose toxicity studies showed that the BMVs obtained from Δ*msbB*
*E. coli* did not cause any mortality even at a single high dose of 100 µg, while, on the other hand, administration of 50 µg of BMVs obtained from wild type *E. coli* lead to mortality within 48 h post administration. This shows how the biochemical composition of the surface moieties of BMVs play a crucial role in its toxicity response. Here, the Δ*msbB* mutation in *E. coli* produces underacylated LPS which shows reduced endotoxicity.

While BMVs show great promise as a drug delivery vehicle, one report has gone further to evaluate its potential as a stimuli-responsive multifunctional theranostic agent [91]. To do so, BMVs synthesized from Δ*msbB*
*E. coli* were loaded with melanin (which can act both as a photoacoustic and photothermal agent), by introducing the required plasmid to the bacteria. This led to the generation of melanin which is packed within the BMV lumen. Upon laser exposure, a concentration dependent thermal response was observed for melanin loaded BMVs, which when used in vitro produced significant cell death due to the photothermal effect (Figure 14A,B). When such melanin-loaded BMVs were administered in 4T1 mammary tumor-bearing FOX-N-1 nude mice intravenously at a single dose of ~150 µg protein, optoacoustic signals could be observed in tumor, liver, and kidneys, enabling the study of its biodistribution profile. The imaging results demonstrated that these BMVs accumulated in the tumor through EPR effects, underwent continuous circulation in vivo, and cleared slowly from the system over a period of 24 h. Photothermal treatment of the 4T1 tumors 3 h post-injection of a single dose of ~75 µg BMVs lead to a significant thermal response of 56 °C and 47 °C for intratumoral and intravenous administration respectively. This resulted in a significant reduction in tumor volume over an 8 days period after just a single treatment and laser therapy which its high effectiveness in cancer therapy (Figure 14C,D). As previous studies have pointed out, there occurred a significant increase in the cytokine levels of TNF-α, IL-6, and IFN-γ 2 h post administration of BMVs, which however reduced near to the baseline levels after 25 h.

### 7.2. Antibacterial Therapy

The use of BMVs for antibacterial applications has mostly been investigated by utilizing the inherent antigenic molecules present on BMV surface for eliciting a favorable immune response against the invading pathogen. In this regard, BMVs have been utilized either directly in its unmodified form [80] or modified appropriately to present the antigenic proteins with a suitable material to elicit a desired immune response. One way of presenting antigenic proteins to the immune system is through the use of NPs, which can maximize immune cell recognition owing to their large surface area and size scale that facilitates particle uptake [115,116,117]. Numerous literature reports have utilized this methodology to develop nanotechnology-based vaccines and similar approaches have been demonstrated with BMVs for antibacterial therapy.

To demonstrate the potential of NP-based immunomodulation by utilizing BMV antigen proteins, surface-modified Au NPs have been investigated in vivo for antibacterial therapy applications [87]. Upon intravenous administration of BMV protein-coated Au NPs (2.5 µg dose) into immunocompetent CD-1 mice, a number of highly precise immune responses were observed as compared to bare BMVs and PBS injected controls. Specifically, it was observed that BMV-coated Au NPs lead to a heighted activation and maturation of dendritic cells and T cells and an increased B cell response and a consequent increase in antibody titers. Note, in this study, that when smaller sized Au NPs (~30 nm) were used as compared to larger sized Au NPs (~90 nm) a relatively heighted dendritic cell activation and maturation was observed. This was found to be due to the better accumulation of smaller sized Au NPs in the lymph nodes of mice, thereby making them more suitable for immune activation. However, in this study the application of these BMV protein surface-modified Au NPs was not evaluated in an infection model.

In an interesting report, BMVs were utilized to surface modify polymeric NP-based antibiotic delivery systems such that target specificity could be achieved [84]. Preliminary in vitro studies undertaken by this group showed that BMVs isolated either from *E. coli* or *S. aureus* showed specific uptake in macrophages that were pre-infected with *E. coli* or *S. aureus*, respectively. As a result, antibiotic-loaded PLGA NPs surface modified with *S. aureus* BMVs were found to show significantly greater accumulation in major organs of an *S. aureus* infected BALB/c mice model as compared to *E. coli* BMV-coated PLGA NPs and liposome-coated PLGA NPs. This greater internalization was explained to be caused by the greater ability of infected macrophages to internalize the NPs first, followed by the natural biodistribution of these macrophages to the infected organs. As a result of this greater accumulation observed for the *S. aureus* BMV-coated antibiotic loaded PLGA NPs, effective reduction of bacterial CFU counts could be observed in kidneys and lungs of the infected mice. However, note here that effective therapeutic efficacy could not be achieved for all major organs, and more studies need to be carried out to exploit this specific property of BMV-coated NPs. It would be interesting to study in this case if the specific uptake of BMV-coated NPs in infected macrophages could be extended to other infected mammalian cells and cancer cells. If such a specificity could be achieved, it could open up the possibility of treating patients with tumors bearing bacterial load. Such infected tumor conditions are nowadays observed in clinical investigations and the presence of these bacteria in tumors are found to play an important role in inhibiting the efficacy of chemotherapeutic drug treatments [118,119].

Beyond the use of BMVs, one unique report demonstrated the direct use of bacterial cells for NP surface modification [120]. Here, bacterial protoplasts were first isolated by treating bacteria with lysozyme to remove the bacterial cell wall, followed by its serial extrusion through 10, 5, and 1 µm sized polycarbonate membrane filters. These protoplast-derived nanovesicles (PDNVs) show inherent advantage over BMVs as they can be directly synthesized in large amounts from the bacterial suspension. Additionally, due to removal of the bacterial cell wall, the resultant PDNVs were depleted of the outer membrane proteins OmpA and lipid A which are components of LPS, thus making them less immunogenic and more favorable for drug delivery and theranostic applications. Here, for investigating its immunotherapeutic potential, PDNVs were loaded separately either with *E. coli* antigen OmpA and *S. aureus* antigen Scoagulase by expressing the desired antigen in the parent *E. coli* bacterium. The resultant PDNVs, when administered in vivo in C57BL/6 mice, showed high specificity in developing an immune-protective response against bacterial challenge. Specifically, it was observed that PDNVs harboring either the OmpA antigen or the Scoagulase antigen showed effective immune response in vivo when challenged with lethal doses of *E. coli* or *S. aureus* respectively. This response was found to be highly specific with respect to the antigen that the PDNVs harbored such that mice that were administered with PDNVs harbouring *E. coli* antigen OmpA succumbed when challenged with *S. aureus* infection, and vice versa. The protective immunity offered by such PDNVs could be observed in mice up to 6 weeks post-administration. Interestingly, these PDNVs were found to show low in vitro and in vivo toxicity as compared to *E. coli*-derived BMVs. Specifically, it was observed that even upon administration of upto 1 mg of PDNVs in C57BL/6 mice all animals survived, while a dosage of only 25 µg of BMVs lead to the death of 80% of the animals. Overall, PDNVs show great promise towards the development of biomimetic drug delivery vehicles and show several advantages as compared to BMVs.

## 8. Challenges and Future Perspective of Utilizing BMVs for Drug Delivery

The use of BMVs for drug delivery applications would encounter challenges that are similar to the design and synthesis of other drug delivery systems as well as other unique challenges that pertain to it alone. One of the most important properties of drug delivery systems over and above its therapeutic efficacy is its safety and immunogenicity. Unlike in the use of BMVs for vaccine development, for drug delivery applications, it is imperative that BMVs do not elicit an immunogenic response in the body. The presence of immunogenic molecules such as LPS and LTA and other bacterial proteins can impede the utility of BMVs for such applications. In such cases, efforts need to be focused on utilizing and developing bacterial strains that can produce BMVs containing lower amounts of or attenuated immunogenic molecules (e.g., Δ*msbB*
*E. coli* [83]). To prevent immune recognition other methods such as pegylation [121] or the incorporation of anti-phagocytic CD-47 molecules [122] on the surface of BMVs could be utilized. However, this could lead to additional synthesis/modification steps which could add to the complexity of the otherwise simple BMV system.

Another challenge that researchers face at this stage in utilizing BMVs for drug delivery applications is the lack of knowledge regarding BMV synthesis routes, mechanism of cargo packaging, and factors affecting the same. Knowledge regarding these can enable the design of suitable strategies for maximizing vesicle synthesis and cargo loading such that cargo loaded BMVs could be obtained in the first instance of BMV synthesis. This is advantageous because separation and purification procedures would be comparatively simple for preloaded BMVs as compared to post-loading strategies (after BMV separation) wherein unloaded cargo, empty BMVs, and cargo-loaded BMVs have to be separated individually. To overcome challenges related to separation, affinity-based separation could be utilized if the cargo of interest is also coupled with a desired protein tag. However, in this case the cargo and protein tag should be present on the BMV surface such that the proteins face the outer surface of the vesicles in order to interact with the affinity molecule used for separation. Magnetic separation of BMVs could also be a viable alternative if the cargo of interest is coupled to magnetic NPs which can then be loaded into BMVs. Such a separation technique has been demonstrated successfully for exosomes [123].

As the use of BMVs for drug delivery applications is still in its infancy, and BMVs have very high heterogeneity (based on the parent bacterial source and strain and growth conditions), it is absolutely essential that researchers working on this field quantify data regarding BMV synthesis, cargo loading, and culture conditions such that comparative analysis across various studies could be made effectively These standard parameters should also include details such as number of BMVs obtained/CFU, isolation protocol, storge conditions, and their effect on long-term stability, protein content, and concentration etc. For drug delivery applications particularly, it is essential that %cargo loading be evaluated and the cargo loading method be detailed. Additionally, it is also important to carry out studies on evaluating the differences on utilizing BMVs of different sizes for drug delivery obtained from the same bacterial source to understand if the size parameter plays an important role in their overall cargo delivery/pharmacokinetics. Understanding and cataloging such information would help steer and accelerate the use of BMVs for drug delivery applications in the future.

From the engineering standpoint, one major challenge that BMVs would face for their utility for biomedical application is their difficulty in separation and purification for clinical translation. At present, a number of different steps are required to separate BMVs from the parent bacterial growth culture and other extraneous proteins, which in the long-run can increase their cost of production. Other alternative low-energy separation techniques need to be developed and optimized for successful translation of BMVs for drug delivery. Ultimately, the cost-to-benefit ratio of synthesizing and purifying BMVs from bacterial culture must be assessed as compared to other promising biomimetic systems such as exosomes which could require a large-scale mammalian culture facility.

## 9. Conclusions

Even though the use of BMVs for drug delivery applications is still in its infancy, it has tremendous potential to become a successfully translatable drug delivery system. This is primarily because of their ability to be produced innately by bacteria in large quantities using inexpensive medium and culture conditions. Until now, BMVs have been investigated for the delivery of a number of different cargos including chemotherapeutic drugs, therapeutic nucleic acids, antibiotics, NPs, etc. and have been evaluated in vivo in a few studies with promising results. Nevertheless, more thorough and detailed investigations are required on evaluating BMVs for different biomedical applications.

## Figures and Tables

**Figure 1 pharmaceutics-13-01430-f001:**
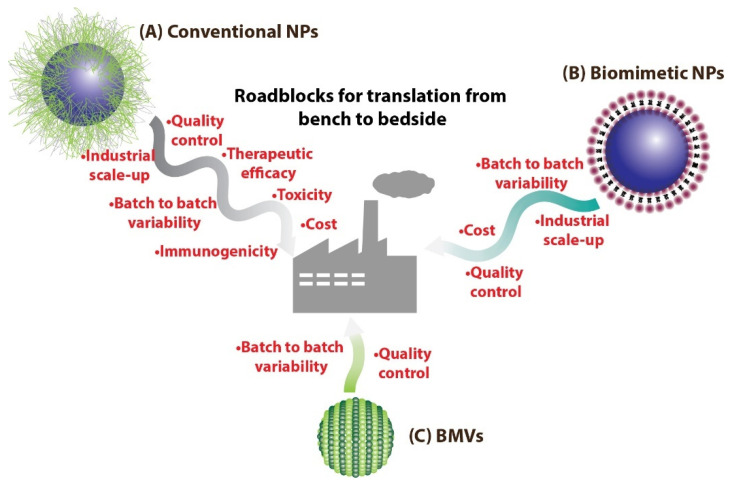
(**A**) Conventional NPs synthesized routinely in laboratories could face many more roadblocks on the path to clinical translation as compared to (**B**) biomimetic NPs. On the other hand, (**C**) BMV-based nanomedical systems could benefit over other biomimetic NPs as they have the potential to be easily mass produced.

**Figure 2 pharmaceutics-13-01430-f002:**
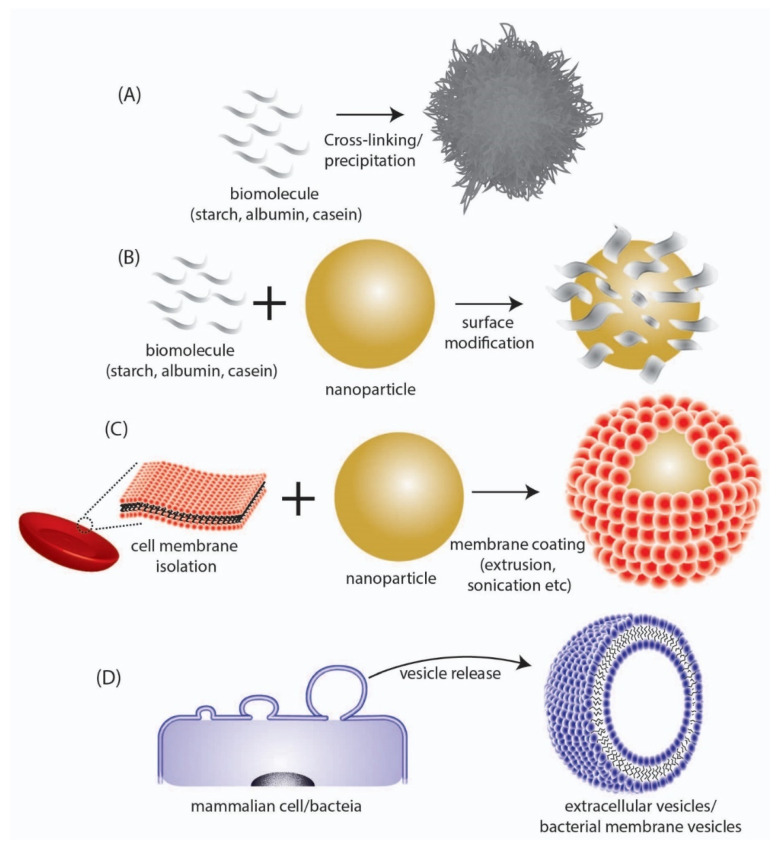
Depiction of different types of biomimetic/bioinspired NPs that have been reported in literature. (**A**) NPs synthesized directly from a biomolecule. (**B**) Surface modification of synthesized NPs with biomolecular structures. (**C**) Coating of synthesized NPs with cell membrane surfaces derived from mammalian cells. (**D**) Direct utilization of nano-sized extracellular vesicles/bacterial membrane vesicles isolated from mammalian or bacteria cells respectively.

**Figure 3 pharmaceutics-13-01430-f003:**
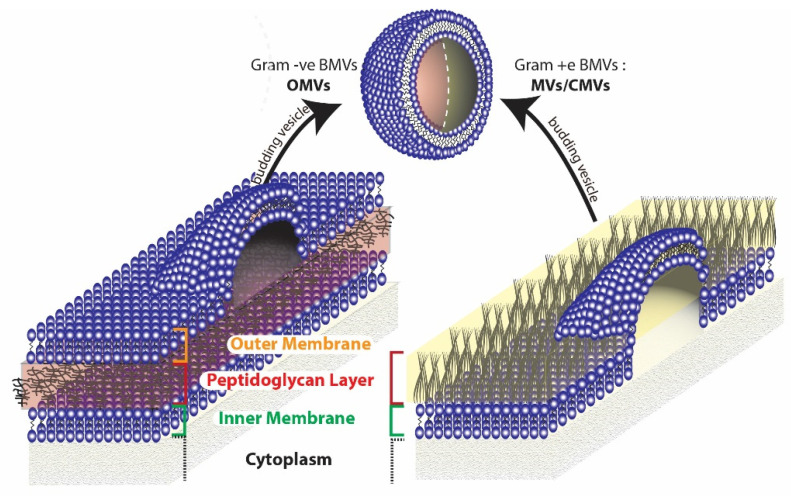
Origin of BMVs differ in gram negative and gram positive bacteria due to the inherent differences in the cell membrane structure.

**Figure 4 pharmaceutics-13-01430-f004:**
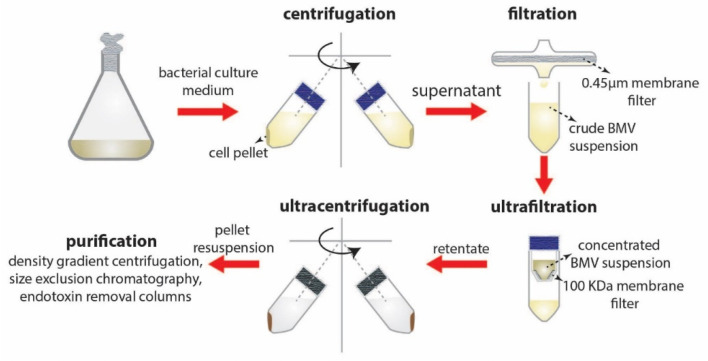
Separation and purification steps involved in BMV isolation.

**Figure 5 pharmaceutics-13-01430-f005:**
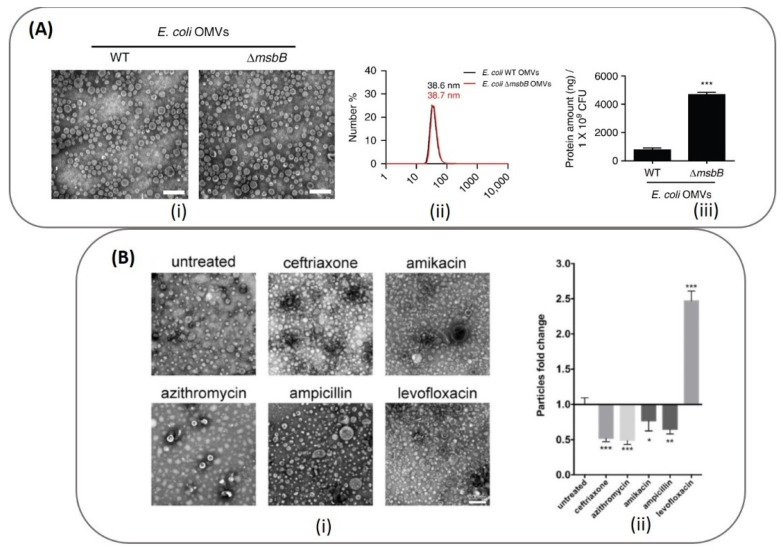
Top panel (**A**): (i) TEM and (ii) DLS data depicting the actual site and hydrodynamic diameter of the OMVs obtained from *E. coli* and its Δ*msbB* mutant (***: *p* < 0.001). The comparative yield of OMVs from each strain was quantified through protein concentration measurements. Reproduced from [83], Springer Nature, 2017. Bottom panel (**B**): (i) TEM image of OMVs obtained from *A. baumannii* cultured in different antibiotics and their (ii) comparative yield measured using NTA analysis (*** *p* < 0.001; ** *p* < 0.01; * *p* < 0.05). Reproduced from [71], Elsevier, 2020.

**Figure 6 pharmaceutics-13-01430-f006:**
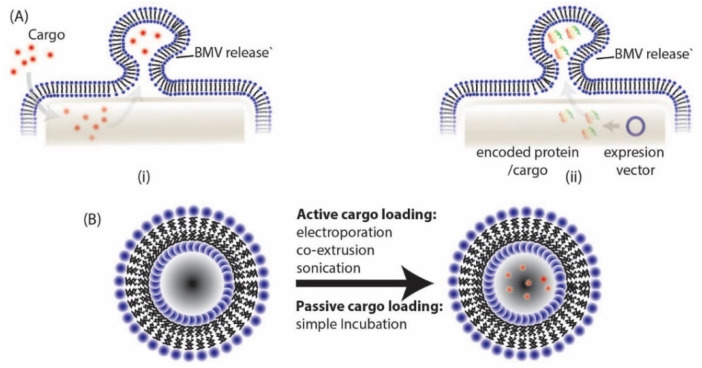
BMVs can be loaded either through (**A**) (i) direct exposure of cargo to parent bacteria, (ii) by transformation of parent bacteria with desired expression vector or (**B**) through different active and passive cargo loading techniques post BMV isolation.

**Figure 7 pharmaceutics-13-01430-f007:**
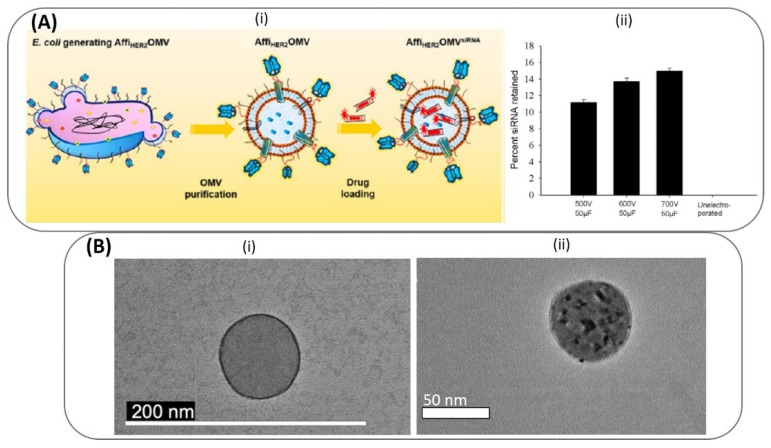
Top panel (**A**): (i) Schematic of electroporation mediated siRNA loading of siRNA into *E. coli* OMVs. (ii) siRNA loading was optimized at different high-voltage pulses. Reproduced from [86], American Chemical Society, 2014. Bottom panel (**B**): (i) TEM image of *P. Aeruginosa* BMV before (i) and after (ii) loading of Au NPs through electroporation. Reproduced from [93], Springer Nature, 2019.

**Figure 8 pharmaceutics-13-01430-f008:**
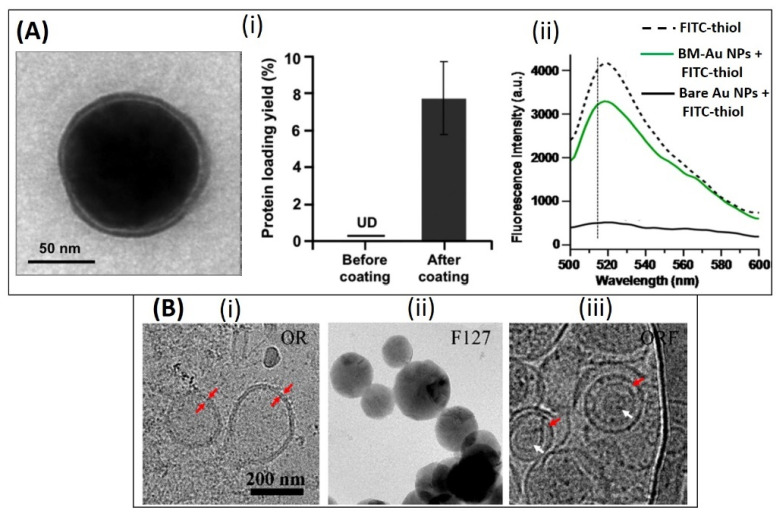
Top panel: (**A**) (i) TEM image of bacterial membrane coated Au NPs and measurement of protein content on Au NPs before and after coating. (ii) Fluorescence quenching assay showing the reduction in fluorescence intensity of FITC-thiol only when incubated with bare Au NPs as compared to bacterial membrane coated Au NPs (BM-Au NPs). Reproduced from [87], American Chemical Society, 2015. Bottom panel: (**B**) TEM image of (i) RGD functionalized OMV [OR], (ii) F127 polymeric micelle and (iii) OMV coated F127 micelle. Reproduced from [82], American Chemical Society, 2020.

**Figure 9 pharmaceutics-13-01430-f009:**
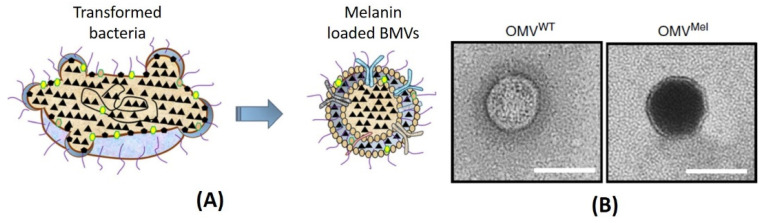
(**A**) Simple schematic showing the loading of melanin from transformed bacteria. (**B**) Comparative of TEM images of unloaded wild type OMV (OMV^WT^) and melanin-loaded OMVs (OMV^Mel^) from *E. coli*. Reproduced with permission from [91], Springer Nature, 2019.

**Figure 10 pharmaceutics-13-01430-f010:**
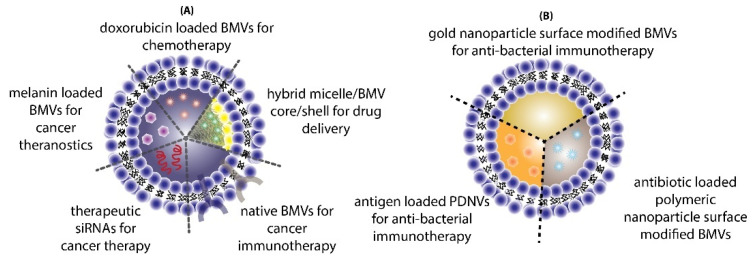
Different applications of BMVs reported in literature for (**A**) cancer therapy and (**B**) antibacterial therapy.

**Figure 11 pharmaceutics-13-01430-f011:**
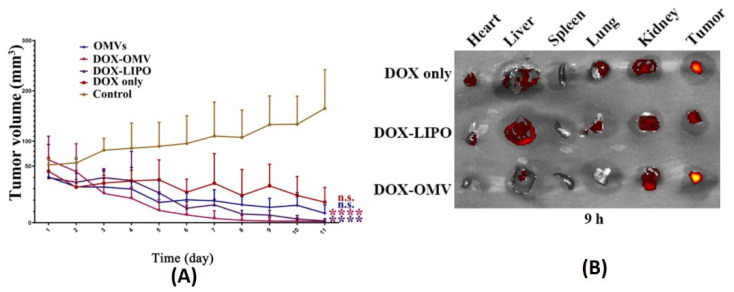
(**A**) Tumor volume reduction measurements and (**B**) in vivo drug distribution of doxorubicin-loaded *K. pneumoniae* OMVs as compared to controls. Reproduced from [81], Elsevier, 2020. **** *p* < 0.001 vs. control, n.s. no statistic difference vs. control.

**Figure 12 pharmaceutics-13-01430-f012:**
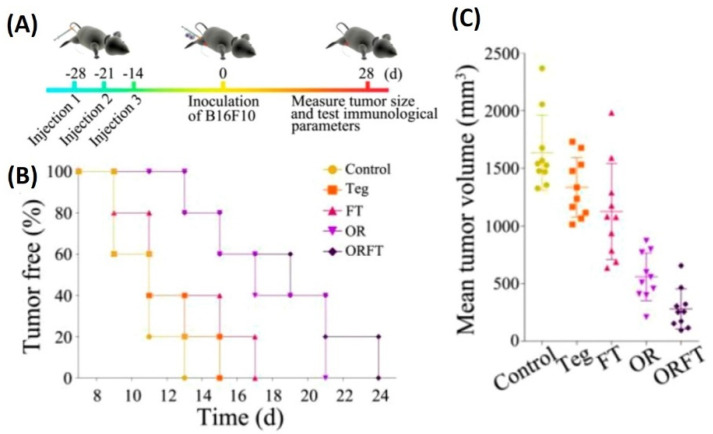
(**A**) Treatment timeline for evaluating the protective role of hybrid BMVs (drug-loaded micelles surface modified with BMVs)—hybrid BMVs were administered to mice before tumor challenge. (**B**) Percentage of tumor-free mice after tumor challenge (**C**) Mean tumor volume measurements of hybrid OMVs against controls (Teg-Tegafur, FT-Tegafur loaded F127 micelle, OR-RGD functionalized BMV, ORFT—Tegafur-loaded F127 micelle surface modified with RGD functionalized BMV). Reproduced with permission from [82], American Chemical Society, 2020.

**Figure 13 pharmaceutics-13-01430-f013:**
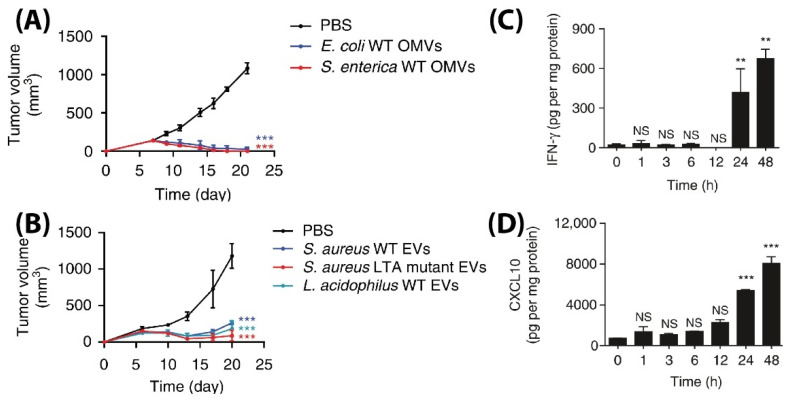
Tumor volume measurement study for CT26 tumors injected with (**A**) Gram-positive *S. Aureus*, lipoteichoic acid *S. Aureus* mutant, and *L. Acidophilus*. (**B**) Gram-negative *E. coli* and *S. enterica*. Measurement of (**C**) IFN-γ and (**D**) CXCL10 cytokine levels in tumor cell lysate at different time points after *E. coli* BMV administration in CT26 tumors (** *p* < 0.01, and *** *p* < 0.001). Reproduced with permission from [83], Springer Nature, 2017.

**Figure 14 pharmaceutics-13-01430-f014:**
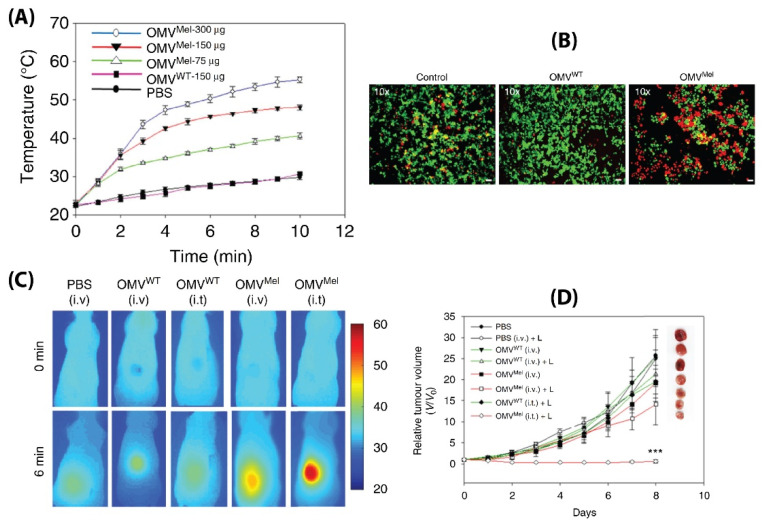
(**A**) Thermal response of melanin-loaded OMVs at different concentrations as compared to wild type OMVs upon laser exposure. (**B**) Live–dead (calcein AM/EthD-1) staining of 4T1 cancer cells treated with melanin-loaded OMVs and exposed to laser as compared to controls. Thermal imaging (**C**) and tumor volume reduction measurements (**D**) of 4T1 tumors after either intravenous or intratumoral administration of melanin loaded OMVs and laser exposure as compared to controls. Reproduced with permission from [91], Springer Nature, 2019. *** *p* < 0.001 vs. PBS with laser treatment.

**Table 1 pharmaceutics-13-01430-t001:** Different types of biomimetic/bioinspired NPs reported in literature.

Type	Biological Source	Cargo Loaded/NP	Application	Reference
Biomolecule assembly	Human Serum albumin	Indocyanine green	Active targeting and photothermal therapy of NIH-3T6.7 tumor (in vivo)	[31]
	Casein	10-hydroxycamptothecin	Drug delivery to C6 glioma tumor (in vivo)	[32]
	Human transferrin	Near-infrared dye IR-780	Photodynamic and photothermal therapy of CT26 colon carcinoma (in vivo)	[33]
	Human H-ferritin	Doxorubicin	Drug delivery to U87MG human glioma	[34]
Surface modification	Bovine serum albumin	Silver NPs	Photothermal ablation of B16F10 murine melanoma (in vitro)	[35]
	Casein	Iron-Oxide NPs	Active EGFR targeting (in vitro) and MRI contrast (in vivo)	[36]
	High density lipoprotein	gold NPs	Nucleic acid delivery to PC3 prostate cancer cells (in vitro)	[37]
Mammalian cell membrane-coated NPs	Erythrocytes	poly(lactic-coglycolic) acid NPs	Toxin removal- demonstrated in mouse sepsis model	[22]
	Neural stem cells	poly(lactic-coglycolic) acid NPs	Glyburide delivery for stroke treatment (in vivo)	[38]
	Platelets	poly(lactic-coglycolic) acid NPs	Rapamycin delivery for atherosclerosis treatment (in vivo)	[26]
	Mouse leukemia cell C1498	poly(lactic-coglycolic) acid NPs	Active targeting and delivery of dexamethasone del for treatment of lung infection (in vivo)	[39]

**Table 2 pharmaceutics-13-01430-t002:** BMVs isolated from different sources utilized for various applications.

Bacterial Species	BMV Size (nm)	Cargo Loaded	Loading Method	Application
*E. coli* K-12 W3110 strain [86]	30–250	siRNA	Electroporation	Anti-tumor therapy
*E. coli* [88]	55 ± 1	NanoLuc Luciferase enzyme	Genetic engineering of parent bacteria	Bioluminescence Imaging
*E. coli* strain BL21 [90]	136 ± 67	Phosphotriesterase enzyme	Genetic engineering of parent bacteria	Environmental remediation
*A. baumannii* [71]	200–300	Antibiotics (ceftriaxone, amikacin, azithromycin, ampicillin, levofloxacin, ciprofloxacin, norfloxacin)	Antibiotic treatment of parent bacteria	Antibacterial Therapy
*E. coli* K-12 W3110 strain [91]	20–200	Melanin	Genetic engineering of parent bacteria	Cancer theranostics
*E. coli* JC8031 [89]	40	NanoLuc Luciferase enzyme	Genetic engineering of parent bacteria	Ability to decorate multiple functional protein moieties demonstrated
*K. pneumoniae* (attenuated) [81]	~70	Doxorubicin	Simple incubation of drug with BMVs	Anti-tumor therapy (drug + immunotherapy)
*P. aeruginosa* [93]	30–200	Gold NPs	Electroporation	Showed ability to load gold NPs in BMV lumen

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
