# Peer review of "Biomimetic Bacterial Membrane Vesicles for Drug Delivery Applications"

_pharmaceutics, 2021, doi:10.3390/pharmaceutics13091430_

Round 1
Reviewer 1 Report
In this manuscript, authors reviewed the advances about biomimetic bacterial membrane vesicles for drug delivery system. In my opinion, some issues should be further address and I hope the following comments could be helpful for improving their paper.
- In the introduction, the background about the bacterial membrane based biomimetic technology is very little, the authors should enrich this part by citing some recent literatures and emphasize the necessity of the biomimetic strategy.
https://onlinelibrary.wiley.com/doi/abs/10.1002/adhm.202002081
- Replace Bio-mimetic with biomimetic in the whole manuscript.
- I am confused about Figure 1, the meaning of this figure is not clear, what does authors want to represent here? Is this figure drawn by himself? If it is yes, then why there is reference in the figure legend?
- I am wondering, how drug will release from bacterial membrane coated Nps? How will this coat rupture and drug will release?
- Figure 3: Please modify it, it is not clear ? what were written on it?
- Section: 5, this part need more improvement, authors need to put in some recent examples with figures.
- Section 6. Cargo loading and surface modification using BMVs: I didn't notice about surface modification information in this section, authors need to add another new subsection about surface decoration or surface modification of bacterial membrane and add new and recent examples with figures and to tell why surface modification is important, also add one table which summarizes this subsection in simple way.
- This manuscript is well organized but lack of specific comparative analysis. What are the advantages of biomimetic bacterial nanotechnology compared with traditional nanotechnology? Authors need to explain it and also include it in the manuscript.
- The authors should summarize the current approaches of fabricate bacterial cell-based biomimetic nanoparticles and compare their advantages and disadvantages in order to draw the reader's attention.
- In the parts of conclusions and perspectives, the author should consider giving some methodological design to improve the performance of such materials.
- Section 7. Drug delivery applications of BMVs: This part must be improved by adding more recent examples from latest literatures. And it should be put in some more figures with examples along with data shown in the results. It will draw reader 's attention.
- The quality of figures is most important for review paper, authors need to provide high resolution figures.
- The manuscript is not well written. There are a number of misused words. I strongly recommend the authors work with a native English speaker to work through quite a few poorly worded and confusing sentences.

Author Response
Thank you for your valuable review comments.
We improved the quality of the manuscript based on your comments.
Please see the attachment.

Reviewer 2 Report
This is an interesting review of the uses of bacterial membrane vesicles (BMV) as drug delivery agents. The following important points should be clarified, however, before acceptance:
- When the authors refer to “nanoparticles” they have to make clear that nanovesicles (such as liposomes or other type of vesicles, encompassed in the broad definition of nanoparticle) are excluded.
- The “engineering” labels in figure 1 are redundant. Please modify it.
- It seems to me that despite the feasibility of growing massive amounts of bacteria, the subsequent steps of isolation and purification of BMV may not be easily accomplished. Hence, a comparison of production methods milestones [at a lab and industrial scale] between BMV and other already in the market nanoparticles (polymeric and vesicular) would aid to provide a broader landscape to the novel interested reader.
- it is not clear to me if there are already other commercial products based on BMV [Row 408: vaccine against Neisseria meningitidis serogroup B (Bexsero® developed by Novartis)?]. A table depicting current clinical trials (if performed) would be of aid. Otherwise, it must be explicitly stated that for the moment the use of BMV as drug delivery systems is limited to pre-clinics.
- Regarding point 4: what would be, according to authors’ criteria, the requirements for clinical approval of BMV as drug delivery systems, compared to those for liposomes?
- Row 234: please indicate the meaning of the asterisk: “ 4*1011 BMV particles” .
- Rows 417-617: The authors highlight the review will be referred to the use of BMV in drug delivery. However, in almost all the cited examples, the immunomodulatory role of BML remained present. Hence, perhaps the authors should reformulate the way the uses of BML are presented, stating that a combination of activities is provided, which may finally be advantageous compared to isolated immunomodulation and chemotherapy.

Author Response

(The authors gave the same response as above.)

Round 2
Reviewer 1 Report
Authors addressed all comments. I am accepting this manuscript in current form.